

# Whole genome sequencing of *Streptococcus pneumoniae*: development, evaluation and verification of targets for serogroup and serotype prediction using an automated pipeline

Georgia Kapatai[1,*], Carmen L. Sheppard[1,*], Ali Al-Shahib[2], David J. Litt[1], Anthony P. Underwood[2], Timothy G. Harrison[1] and Norman K. Fry[1]

[1] Respiratory and Vaccine Preventable Bacterial Reference Unit, Public Health England, London, United Kingdom
[2] Infectious Disease Informatics, Public Health England, London, United Kingdom
[*] These authors contributed equally to this work.

Corresponding author
Georgia Kapatai,
Georgia.Kapatai@phe.gov.uk

## ABSTRACT

*Streptococcus pneumoniae* typically express one of 92 serologically distinct capsule polysaccharide (cps) types (serotypes). Some of these serotypes are closely related to each other; using the commercially available typing antisera, these are assigned to common serogroups containing types that show cross-reactivity. In this serotyping scheme, factor antisera are used to allocate serotypes within a serogroup, based on patterns of reactions. This serotyping method is technically demanding, requires considerable experience and the reading of the results can be subjective. This study describes the analysis of the *S. pneumoniae* capsular operon genetic sequence to determine serotype distinguishing features and the development, evaluation and verification of an automated whole genome sequence (WGS)-based serotyping bioinformatics tool, PneumoCaT (**Pneumo**coccal **Ca**psule **T**yping). Initially, WGS data from 871 *S. pneumoniae* isolates were mapped to reference cps locus sequences for the 92 serotypes. Thirty-two of 92 serotypes could be unambiguously identified based on sequence similarities within the cps operon. The remaining 60 were allocated to one of 20 'genogroups' that broadly correspond to the immunologically defined serogroups. By comparing the cps reference sequences for each genogroup, unique molecular differences were determined for serotypes within 18 of the 20 genogroups and verified using the set of 871 isolates. This information was used to design a decision-tree style algorithm within the PneumoCaT bioinformatics tool to predict to serotype level for 89/94 (92 + 2 molecular types/subtypes) from WGS data and to serogroup level for serogroups 24 and 32, which currently comprise 2.1% of UK referred, invasive isolates submitted to the National Reference Laboratory (NRL), Public Health England (June 2014–July 2015). PneumoCaT was evaluated with an internal validation set of 2065 UK isolates covering 72/92 serotypes, including 19 non-typeable isolates and an external validation set of 2964 isolates from Thailand ($n = 2,531$), USA ($n = 181$) and Iceland ($n = 252$). PneumoCaT was able to predict serotype in 99.1% of the typeable UK isolates and in 99.0% of the non-UK isolates. Concordance was evaluated in UK isolates where further investigation was possible; in 91.5% of the cases the predicted capsular type was
concordant with the serologically derived serotype. Following retesting, concordance increased to 99.3% and in most resolved cases (97.8%; 135/138) discordance was shown to be caused by errors in original serotyping. Replicate testing demonstrated that PneumoCaT gave 100% reproducibility of the predicted serotype result. In summary, we have developed a WGS-based serotyping method that can predict capsular type to serotype level for 89/94 serotypes and to serogroup level for the remaining four. This approach could be integrated into routine typing workflows in reference laboratories, reducing the need for phenotypic immunological testing.

## INTRODUCTION

*Streptococcus pneumoniae* is a human respiratory tract pathogen that represents a leading cause of invasive bacterial disease in children under five years of age and the elderly. Isolates of *S. pneumoniae* are traditionally characterised in terms of the antigenicity of their capsular polysaccharides (CPS) and currently there are 92 serologically distinct serotypes as defined by Statens Serum Institut (SSI), Copenhagen, Denmark sera using the Neufeld reaction for capsular swelling (referred to as Quellung) (*Austrian, 1976*; *Statens Serum Institut, 2013*). Additional 'serotypes' (6E, 6F, 6G, 6H, 11E, 20A and 20B) have been identified by molecular methods and/or monoclonal antibodies, but these are not distinguishable using the current commercial serotyping sera and some may not be phenotypically distinct types, but simply genetic variants producing the same polysaccharide structures (*Calix & Nahm, 2010*; *Calix et al., 2012*; *Ko, Baek & Song, 2013*; *Oliver et al., 2013a*; *Park et al., 2015*; *Burton et al., 2016*). The polysaccharide capsule is also the target of all currently licensed vaccines and therefore, due to selective pressure on circulating strains, accurate identification of pneumococcal serotypes is essential for disease surveillance, evaluation of the efficacy of the pneumococcal vaccines and to inform national vaccine policy.

In the UK, the initial reduction of invasive pneumococcal disease (IPD) and carriage, due to the introduction of PCV7 and PCV13 polysaccharide-conjugate vaccines, was followed by increases in non-vaccine type IPD (*Miller et al., 2011*; *Waight et al., 2015*). Specifically, following implementation of pneumococcal conjugate vaccine PCV13 (*Centers for Disease Control and Prevention (CDC), 2010*) in 2010, changes in disease-causing serotypes included a significant reduction in PCV13 serotypes 1, 6A, 7F and 19A, followed by a substantial increase in non-vaccine serotypes 8, 12F, 15A, 15B/C, 22F, 23B and 24F observed in 2013/14 in children under 5 years of age, when compared with 2012/13 (*Waight et al., 2015*) (Fig. 1). Surveillance studies from other countries revealed a similar pattern of PCV13-serotype reduction and non-vaccine serotype increase in IPD cases. These studies underline the need for continuous surveillance to monitor the emergence of serotypes due to the clonal expansion of non-vaccine serotypes (*Richter et al., 2013*; *Regev-Yochay et al., 2015*).

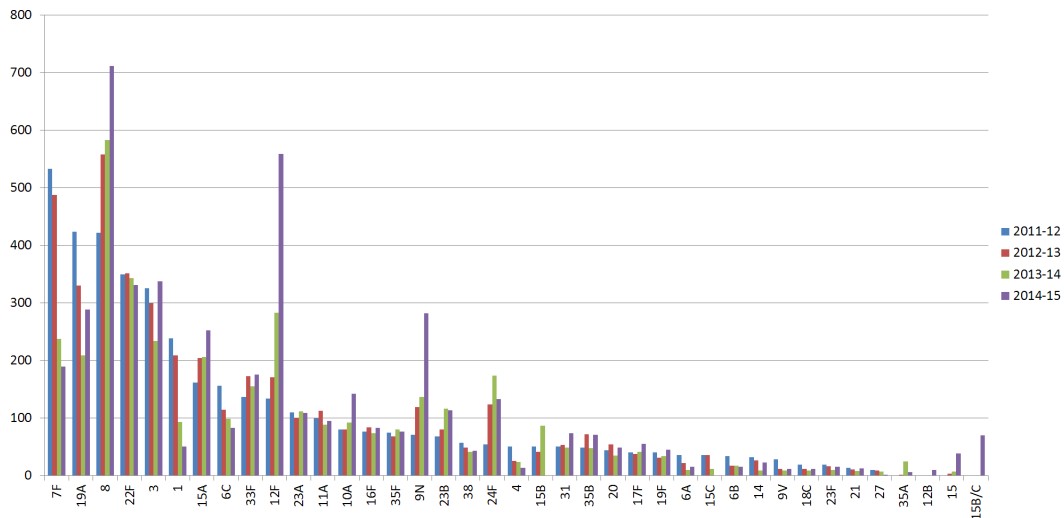

**Figure 1** **Serotype distribution of UK pneumococcal isolates.** Number of IPD isolates referred to PHE per epidemiological year (July 2011–June 2015). Serotypes with less than 10 isolates in all periods were removed.

The Quellung reaction, in conjunction with the antisera from SSI, is currently considered the gold standard for pneumococcal capsular typing and can identify all 92 serotypes (*Austrian, 1976*; *Habib, Porter & Satzke, 2014*). Agglutination tests were developed later to reduce the time required to serotype a batch of isolates by eliminating the need for microscopy. In these tests SSI antisera reacts with the pneumococcal capsule and produces visible clumping (*Kronvall, 1973*; *Porter, Ortika & Satzke, 2014*). Based on the SSI serotype scheme, there are 25 distinct serotypes and 21 serogroups, each consisting of between two and five immunologically-related serotypes, accounting for a further 67 serotypes (*Statens Serum Institut, 2013*). Despite the wide use of the SSI sera, the serological methods can be difficult to interpret since, in certain cases, serotypes within a serogroup can cross-react with some of the antisera making it impossible to resolve to serotype level. In these cases, the serotypes are reported to serogroup level (*Jauneikaite et al., 2015*).

*Bentley et al. (2006)* reported the sequences of the capsular biosynthetic loci of 90 pneumococcal serotypes (serotypes 6C and 6D were not included) and linked the capsular gene sequences to known structural and immunological patterns, thus paving the way for molecular capsular typing. Since then, new molecular serotyping methods have been described (microarray; *Turner et al., 2011*), multiplex PCR (*Brito, Ramirez & Lencastre, 2003*), *in silico* inference from WGS data (*Croucher et al., 2011*; *Everett et al., 2012*; *Metcalf et al., 2016*)), utilising the molecular differences between the capsular loci to determine serotype (*Jauneikaite et al., 2015*). A review of methodologies for capsular typing of *S. pneumoniae* including serological assays, semi-automated molecular tests, and one directly from WGS data (*Liyanapathirana et al., 2014*) was published by *Jauneikaite et al. (2015)*. These molecular/WGS assays have some advantages compared to the serological methods described above: easier interpretation, multiplex capability and in some cases no requirement for culture. However, they still have some limitations, especially if they were

to be used for surveillance. Firstly, due to the genetic similarities of certain serotypes, not all serogroups can be resolved. Secondly, most assays described to date only include the most common serotypes so currently not all serotypes can be detected (*Magomani et al., 2014*). Here we describe genetic capsular locus differences for all but four serotypes and demonstrate the use of these differences in predicting capsular type from WGS data using an automated bioinformatics tool that can be incorporated into routine workflows.

## MATERIALS AND METHODS

### Isolate selection

Reference strains ($n = 91$) for all serotypes (excluding 6D) were acquired from Statens Serum Institut. Reference strain for 6D was kindly provided by The National Institute for Health and Welfare (THL), Finland. A total of 926 clinical isolates were selected from the archives of the Public Health England (PHE) National Reference Lab as a test cohort; for serotypes found to belong to a genogroup, at least 10 isolates were selected where available. Post genomic-sequence data cleansing (to remove repeat isolates from the same patient, mixed cultures, other species and MLST partial profiles or failures) resulted in 871 isolates (Development Set in Table 1). In addition, 2079 prospective or research-related isolates were sequenced as part of the UK validation cohort. This cohort covers 72 of the commonly circulating serotypes (including all vaccine serotypes), and includes prospective isolates received by PHE during 2015, isolates selected as part of research projects and epidemiological investigations (15A ($n = 196$) and 19A ($n = 249$), respectively) and archived isolates for rarer serotypes. Post genomic-sequence data cleansing of this dataset resulted in a total of 2065 isolates (Validation Set in Table 1). All isolates were serotyped on receipt as part of the PHE enhanced surveillance programme using slide agglutination with Statens Serum Institut typing sera.

Genomic data for non-UK isolates were obtained from *Streptococcus pneumoniae* isolate database hosted in BIGSdb (http://pubmlst.org/software/database/bigsdb/) (*Jolley & Maiden, 2010*) and the European Nucleotide Archive (ENA; http://www.ebi.ac.uk/ena). Specifically, three collections were used; a set of 2531 isolates from Thailand initially described by *Chewapreecha et al. (2014)*, an Icelandic panel of 252 serogroup 6 isolates described in *Van Tonder et al. (2015)* and a USA panel of 181 invasive isolates available in ENA as study SRP059723.

### DNA extraction and sequencing

Isolates were grown overnight on horse blood agar (PHE Media Services) with 5% $CO_2$. DNA was extracted from an entire plate of growth for each isolate using the QIAsymphony SP automated instrument (Qiagen) and QIAsymphony DSP DNA Mini Kit, using the manufacturer's recommended tissue extraction protocol for Gram negative bacteria (including a 1 h pre-incubation with proteinase K in ATL buffer and RNAse A treatment). DNA concentrations were measured using the Quant-iT dsDNA Broad-Range Assay Kit (Life Technologies, Paisley, UK) and GloMax® 96 Microplate Luminometer (Promega, Southampton, UK). DNA was sent for whole genome sequencing (WGS) by Illumina sequencing using the PHE Genomic Services and Development Unit (Colindale, London,
**Table 1 Number of isolates from each serotype included in this study.**

| Serotype | Development set | Validation set | Total |
|---|---|---|---|
| 1 | 4 | 41 | 45 |
| 2 | 3 | 9 | 12 |
| 3 | 5 | 44 | 49 |
| 4 | 3 | 43 | 46 |
| 5 | 3 | 41 | 44 |
| 8 | 7 | 70 | 77 |
| 13 | 3 | 20 | 23 |
| 14 | 5 | 43 | 48 |
| 20 | 3 | 42 | 45 |
| 21 | 3 | 21 | 24 |
| 27 | 5 | 23 | 28 |
| 29 | 2 | 21 | 23 |
| 31 | 8 | 22 | 30 |
| 34 | 3 | 22 | 25 |
| 36 | 2 | 5 | 7 |
| 37 | 8 | 22 | 30 |
| 38 | 16 | 23 | 39 |
| 39 | 2 | 2 | 4 |
| 40 | 2 | | 2 |
| 42 | 2 | | 2 |
| 43 | 1 | | 1 |
| 44 | 1 | | 1 |
| 45 | 2 | | 2 |
| 46 | 6 | | 6 |
| 48 | 2 | 5 | 7 |
| 06A | 22 | 41 | 63 |
| 06B | 23 | 43 | 66 |
| 06C | 25 | 22 | 47 |
| 06D | 5 | 2 | 7 |
| 07A | 2 | 5 | 7 |
| 07B | 15 | 4 | 19 |
| 07C | 21 | 24 | 45 |
| 07F | 32 | 40 | 72 |
| 09A | 8 | 6 | 14 |
| 09L | 9 | 2 | 11 |
| 09N | 17 | 44 | 61 |
| 09V | 33 | 45 | 78 |
| 10A | 13 | 44 | 57 |
| 10B | 2 | 7 | 9 |
| 10C | 1 | | 1 |
| 10F | 24 | 22 | 46 |
| 11A | 38 | 44 | 82 |

| Serotype | Development set | Validation set | Total |
|----------|-----------------|----------------|-------|
| 11B | 6 | 2 | 8 |
| 11C | 2 | 4 | 6 |
| 11D | 1 | | 1 |
| 11F | 1 | | 1 |
| 12A | 4 | 2 | 6 |
| 12B | 5 | 24 | 29 |
| 12F | 28 | 44 | 72 |
| 15A | 18 | 196 | 214 |
| 15B | 17 | 41 | 58 |
| 15B/C | | 8 | 8 |
| 15C | 14 | 24 | 38 |
| 15F | 2 | 2 | 4 |
| 16A | 1 | | 1 |
| 16F | 15 | 21 | 36 |
| 17A | 3 | | 3 |
| 17F | 15 | 41 | 56 |
| 18A | 4 | 11 | 15 |
| 18B | 4 | 9 | 13 |
| 18C | 5 | 41 | 46 |
| 18F | 2 | 5 | 7 |
| 19A | 28 | 249 | 277 |
| 19B | 1 | 1 | 2 |
| 19C | 1 | 1 | 2 |
| 19F | 18 | 41 | 59 |
| 22A | 9 | 2 | 11 |
| 22F | 31 | 43 | 74 |
| 23A | 26 | 23 | 49 |
| 23B | 45 | 79 | 124 |
| 23F | 22 | 41 | 63 |
| 24A | 1 | 2 | 3 |
| 24B | 6 | | 6 |
| 24F | 19 | | 19 |
| Serogroup 24 | | 31 | 31 |
| 25A | 3 | 1 | 4 |
| 25F | 10 | 1 | 11 |
| 28A | 15 | 19 | 34 |
| 28F | 4 | 1 | 5 |
| 32A | 2 | | 2 |
| 32F | 3 | | 3 |
| 33A | 3 | 5 | 8 |
| 33B | 4 | 1 | 5 |
| 33C | 2 | 1 | 3 |

**Table 1** (*continued*)

| Serotype | Development set | Validation set | Total |
|----------|-----------------|----------------|-------|
| 33D | 1 | | 1 |
| 33F | 19 | 43 | 62 |
| 35A | 9 | 24 | 33 |
| 35B | 29 | 23 | 52 |
| 35C | 2 | 4 | 6 |
| 35F | 13 | 22 | 35 |
| 41A | 3 | 1 | 4 |
| 41F | 2 | | 2 |
| 47A | 1 | | 1 |
| 47F | 1 | | 1 |
| NT | | 19 | 19 |
| **Grand total** | **871** | **2,067** | **2,938** |

UK) (*Dallman et al., 2014*). Illumina Nextera DNA libraries were constructed and sequenced using the Illumina HiSeq 2500.

Casava 1.8.2 (Illumina inc., San Diego, CA,USA) was used to deplex the samples and FASTQ reads were processed with Trimmomatic (*Bolger, Lohse & Usadel, 2014*) to remove bases from the trailing end that fall below a PHRED score of 30. K-mer identification software (https://github.com/phe-bioinformatics/kmerid) was used to compare the sequence reads with a panel of curated NCBI Refseq genomes to identify the species. A sample of k-mers (DNA sequences of length k) in the sequence data are compared against the k-mers of 1769 reference genomes representing 59 pathogenic genera obtained from RefSeq. The closest percentage match is identified, and provides initial confirmation of the species. This step also identifies samples containing more than one species of bacteria (i.e., mixed cultures) and any bacteria misidentified as *Streptococcus pneumoniae* by the sending laboratory. Further analysis continued only if *S. pneumoniae* was identified. FASTQ reads from all sequences in this study were submitted to ENA using the ena_submission tool (https://github.com/phe-bioinformatics/ena_submission) and can be found at the PHE Pathogens BioProject PRJEB14267 at ENA (http://www.ebi.ac.uk/ena/data/view/PRJEB14267; Tables S1–S2).

## Polysaccharide capsule operon locus sequence analysis

GenBank and FASTA files for the capsular locus sequences for the 90 serotypes published by *Bentley et al. (2006)* were retrieved from NCBI (http://www.ncbi.nlm.nih.gov/) using accession numbers CR931632–CR931722 (*Bentley et al., 2006*). For serotypes 6C and 6D accession numbers JF911515.1 and HV580364.1 were used, respectively. Capsular locus sequences for 6E and 23B1, a new 23B genetic subtype identified during the course of this project, were derived in-house using the assembly-based approach described below. Isolates for both types (6E $n = 12$; 23B1 $n = 27$) were assembled using SPAdes (*Bankevich et al., 2012*) and the capsular locus sequences were extracted using the sequence for 6B and 23B, respectively. The extracted sequences were aligned using ClustalW (*Thompson, Higgins & Gibson, 1994*) and the alignment was visualized with MEGA6 (*Tamura et al., 2013*).

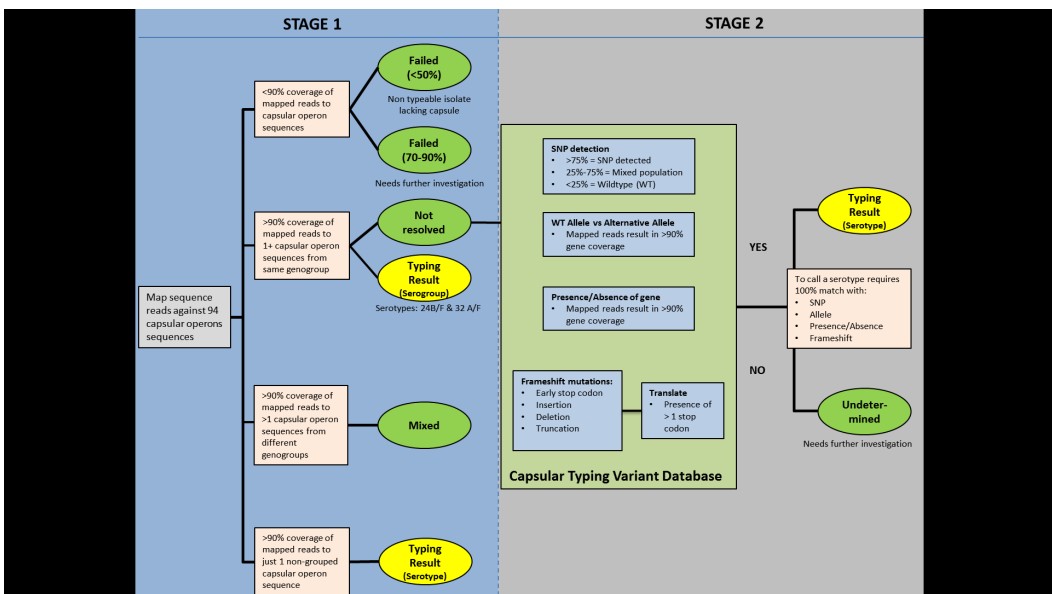

**Figure 2  PneumoCaT process workflow.**

Based on the alignment, consensus sequences were derived for each type and submitted to ENA as assemblies under PHE Pathogens BioProject PRJEB14267 with accession numbers LT594598.1 and LT594599.1 for 6E and 23B1, respectively. Using the GenBank annotation, gene sequences were retrieved from all capsular types. BLAST+ (version 2.2.27) (*Camacho et al., 2009*) alignment was then used to detect gene-level differences within each genogroup.

## Capsular typing tool implementation

PneumoCaT (Pneumococcal Capsule Typing; https://github.com/phe-bioinformatics/PneumoCaT), written in Python (version 2.7.6), utilises a two-step method to assign capsular type (Fig. 2). In the first step, reads from each readset are mapped to a multi-fasta file containing 94 capsular locus sequences (92 serologically distinct serotypes + 2 molecular types/subtypes –6E and 23B1) using bowtie2 (version 2.1.0; following options used: –fr –minins 300 –maxins 1100 -k 99999 -D 20 -R 3 -N 0 -L 20 -I S,1,0.50) (*Langmead & Salzberg, 2012*). Thresholds of coverage of >90% of the length of the sequence, minimum depth of 5 reads per bp and a mean depth of >20 reads over the entire length of the sequence are implemented during this step. This step is considered successful if one or more capsular locus sequences are returned.

If a match to a single capsular locus is returned and the capsular type predicted does not belong to one of our defined genogroups (Table 2), then the software terminates here and reports this as the predicted capsular type (Fig. 3). If the match belongs to a genogroup or more than one locus is matched then the software moves to the second step; a variant-based approach. If multiple serotypes are matched in step 1 and they do not correspond to the same genogroup then a 'Mixed sample' flag is called. Finally, if no serotypes are called due to low coverage (<90%) then a 'Failed' flag is called.

**Table 2** Comparison of genogroups defined in this study and genetic subclusters defined by Mavroidi et al.

| Genogroups | Genetic subclusters |
| --- | --- |
| 6A, 6B, 6C, 6D, 6E | 6A, 6B |
| 7A, 7F | 7A, 7F |
| 7B, 7C, 40 | 7B, 7C, 40 |
| 9A, 9L, 9V, 9N | 9A, 9L, 9V, 9N |
| 10A, 10B | 10A, 10B, 10C, 10F |
| 10C, 10F | |
| 11A, 11B, 11C, 11D, 11F | 11A, 11B, 11C, 11D, 11F |
| 12A, 12B, 12F, 44, 46 | 12A, 12B, 12F, 44, 46 |
| 15A, 15B, 15C, 15F | 15A, 15B, 15C, 15F |
| 18A, 18B, 18C, 18F | 18A, 18B, 18C, 18F |
| 19B, 19C | 19A, 19B, 19C, 19F |
| 22A, 22F | 22A, 22F |
| 23A, 23B, 23F | 23A, 23B, 23F |
| 24A, 24B, 24F | 24A, 24B, 24F, 17F, 48 |
| 25A, 25F, 38 | 25A, 25F, 38 |
| 28A, 28F | 28A, 28F, 16F |
| 32A, 32F | 32A, 32F, 27 |
| 33A, 33F, 37 | 33A, 33F |
| 33B, 33D | 33B, 33D |
| 35A, 35C, 42 | 35A, 35C, 42 |
| 41A, 41F | 41A, 41F, 17A, 31 |

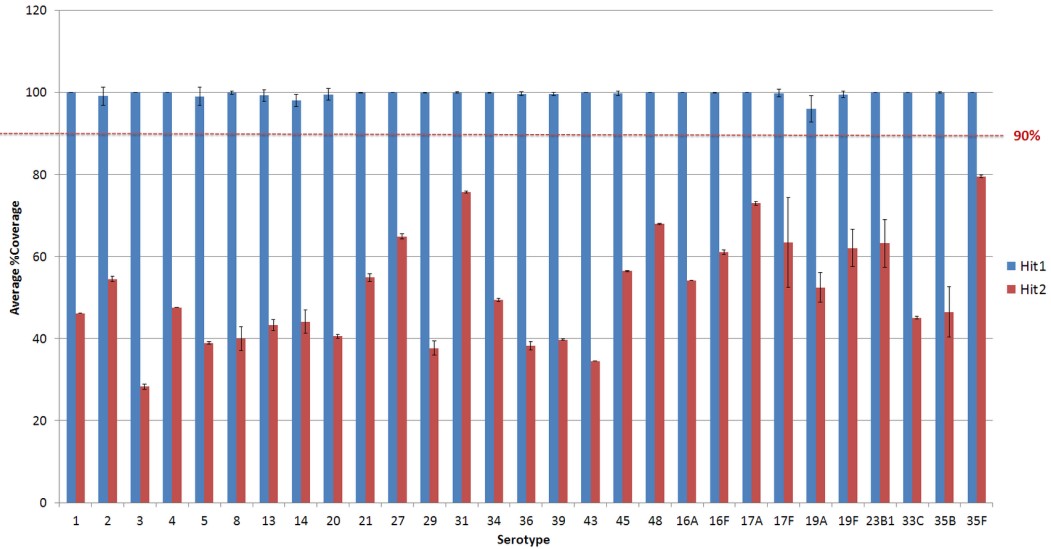

**Figure 3** Average % coverage of the top and second hit during stage 1 of the PneumoCaT for serotypes determined by mapping coverage only. Error bars correspond to standard deviation.

The capsular types matched in step 1 (genogroup) are fed into step 2 and the relevant information for this specific genogroup is retrieved, if genogroup information is available. This information is organised in a multi-fasta reference file, with complete gene sequences for all genes that display genetic diversity within the selected genogroup, and a database containing all sequence variants pertaining to the same genogroup. Paired reads are then mapped to the loaded reference file using bowtie2 and the tool tests for the variants loaded from the capsular type variant (CTV) database. Specific functions were designed to detect the different types of variants, such as presence/absence of genes, different alleles, single nucleotide polymorphisms (SNPs) and loss-of-function mutations. Base calling and insertion/deletion detection were made with SAMtools mpileup (version 0.1.19; following options used: -B -A) (*Li et al., 2009*), requiring a minimum-read depth of 5 and using reads with minimum base mapping quality of 30. These base calls were only included in the downstream analysis if they constituted >80% of read depth (ie, a nonmixed base call). A capsular type is assigned if a 100% match to all variants tested is achieved, else a 'Failed' flag is returned. In the case of serogroups 24 and 32, where no distinguishing variants are available, the software returns the serogroup.

### Phylogenetic tree generation

Reads were mapped to selected reference sequence using bwa (version 0.7.12) (*Li & Durbin, 2009*). Variants were called using GATK 2.6.5 (*McKenna et al., 2010*). Variants were then parsed to retain high quality SNPs based on the following conditions: depth of coverage (DP) $\geq$ 5, AD ratio (ratio between variant base and alternative bases) $\geq$ 0.8, Mapping Quality (MQ) $\geq$ 30, ratio of reads with MQ0 to total number of reads $\leq$ 0.05. All positions that fulfilled the filtering criteria in >0.9 of the samples were joined to produce a multiple fasta format file where the sequence for each strain consists of the concatenated variants. This file was used as an input to generate a maximum likelihood (ML) tree using RAxML (*Stamatakis, 2014*) with the following parameters –m (substitutionModel) ASC_GTRCAT –asc_corr=lewis (Paul Lewis's model correction for ascertainment bias when analysing variant-only data) –b (bootstrapRandomNumberSeed) 12345 -# (numberOfRuns) 1000.

### Assembly-based sequence analysis

Genomic reads were assembled using SPAdes (version 2.5.1) *de novo* assembly software (*Bankevich et al., 2012*) with the following parameters 'spades.py –careful −1 strain.1.fastq.gz −2 strain.2.fastq −t 4 −k 21, 33, 55, 77, 85, 93'. The resulting contigs.fasta file was converted into a BLAST database using blast+ (*Camacho et al., 2009*) (version 2.2.27) and queried using selected query sequence (i.e., gene or capsular operon sequences).

## RESULTS

### Genogroups

Genetic clustering of *S. pneumoniae* strains based on the capsular locus sequence was previously described by *Mavroidi et al. (2007)*. Based on their analysis, 88 pneumococcal serotypes were grouped in eight major clusters and 20 subclusters. All but six serogroups fell within the same genetic subcluster as previously described (*Mavroidi et al., 2007*) and

9 of the subclusters contained serotypes from different serogroups. In this study, we define genogroups as groups of serotypes with high (>90%) sequence identity within the capsular locus. Based on these criteria, 21 genogroups were identified by comparing published capsular locus sequences (*Bentley et al., 2006*). All genogroups contain serotypes grouped within the same genetic subcluster as defined by *Mavroidi et al. (2007)* (Table 2); 11 of the genogroups correspond to serogroups as defined by the SSI sera, six of which also form genetic subclusters described by *Mavroidi et al. (2007)*. Of the remaining 10 genogroups, six are identical to previously described subclusters and three correspond to incomplete serogroups. For example, serogroup 10 assigned previously to a single genetic subcluster is split into two genogroups, 10A–10B and 10C–10F. Similarly, serogroup 19 was assigned to a single subcluster, whereas in our analysis only serotypes 19B and 19C form a genogroup. The remaining two serotypes are too genetically dissimilar to be included; 19A shares only 50% of the capsular locus with 19F, its closest genetic neighbour within the serogroup, whereas 19F shares 65% with 19B. The last genogroup is 33A, 33F and 37; serotypes 33A and 33F belong to a genetic subcluster; serotype 37 was not included in the previous analysis since it does not use the Wzx/Wzy-dependent pathway. However, previous analysis revealed that serotype 37 isolates use the *tts* synthase gene which is located elsewhere in the genome to define serotype, but still contain a defective 33F-like capsular locus (*Llull, Munoz & Lopez, 1999*).

## Serotype-defining sequence variants

Serotype-defining variants were detected following gene-based alignment of published capsular locus sequences (*Bentley et al., 2006*). Sequences were extracted in GenBank format and for each genogroup, gene profiling and sequence variant analysis was performed to determine variant profiles that could predict to serotype level. Types of sequence variation investigated include presence/absence of capsular biosynthesis-related genes, detection of inactivating mutations such as early stop codons and frameshifts (insertions, deletions, truncations), detection of differing alleles of genes that can be distinguished by coverage alone due to low sequence similarity (<80%) and SNP differences that can be detected by variant calling. Reported variants were then evaluated with our test cohort ($n = 871$; Table 2) and variants with 100% incidence rate (found in all isolates of serotype tested) were incorporated into capsular type variant (CTV) database (Table 3). Many of the variants detected during this study correspond to structural and genetic differences reported previously (*Kolkman, Van der Zeijst & Nuijten, 1998*; *Bentley et al., 2006*; *Mavroidi et al., 2007*; *Song, Baek & Ko, 2011*; *Yang et al., 2011*; *Oliver et al., 2013b*).

In two cases of previously reported frameshift mutations (7A and 15C), the mutations were confirmed in our test cohort, but no consensus of the inactivated sequence was detected (20–80% of reads carry frameshift mutation). In 7As, an insertion of a thymine (T) at residue 587 of *wcwD* inactivates the glycosyltransferase (Table 3) leading to loss of the side branch D-Galp-($\beta1-2$)-$\alpha$-D-Galp (*Mavroidi et al., 2007*). During evaluation with a cohort of 34 isolates (7A, $n = 9$; 7F, $n = 25$—Table 2), all 7F isolates carried a wild type (wt) *wcwD* gene sequence. In comparison, 7/9 of the 7A isolates also carried a wt *wcwD* sequence, whereas 2/9 (including the SSI 7A type strain) exhibited a mixed profile at residue 587, with
**Table 3 Genetic differentiation of serotypes within serogroups.** Description of variants present in the CTV database.

| Serogroup | Serotype | Distinguishing genetic features | Functional effect |
|---|---|---|---|
| 6 | 6A/6B and 6C/6D | A > G 583 in *wciP* | Amino acid substitution (Ser195Asn) which results to different rhamnose-ribitol linkages (1 → 3 in 6A/6C and 1 → in 6B/6D) (*Mavroidi et al., 2007*; *Sheppard et al., 2010*; *Baek et al., 2014*) |
| | 6A/6C and 6B/6D and 6E | *wciNα* in 6A and 6B / *wciNβ* in 6C and 6B / *wciNγ* in 6E | Allele *wciNα* encodes for galactosyl-transferase whereas *wciNβ* is 200 bp shorter and encodes for a glycosyl-transferase—consistent with changes in structure (*Park et al., 2007*). WciNγ is a chimeric form of *wciNα* (75%) and *wciNβ* (25%). |
| 7 and serotype 40 | 7A/7F | Frameshift mutation insT 587 in 7A *wcw*D gene | Loss of function of glycosyltransferase leading to loss of side branch for 7A (*Mavroidi et al., 2007*). *"Mixed: ['07A','07F']" result corresponds to 7A phenotype (see 'Results') |
| | 7B/7C/40 | SNPs in *wcwK* (Table S1) | Amino acid changes—*wcwK* encodes for a GT but 7C and 40 structure not known |
| 9 | 9A/9V | Frameshift mutation delG 722 in 9A *wcjE* | Loss of function of O-acetyltranferase leads to differences in acetylation |
| | 9L/9N | SNPs in genes wchA, wcjA, wcjB and wzy (Table S2) | Amino acid changes—*wcjA* and *wcjB* encode for glycosyl-tranferases (GT) and changes in these are consistent with presence of glucose in 9N instead of galactose present in residue 3 of the polysaccharide repeat unit of the other three serotypes (*Mavroidi et al., 2007*) |
| | 9A/9V/9L/9N | presence of an additional O-acetyltransferase encoded by *wcjD* in 9A-9V | Differences in acetylation |
| 10 | 10A/10B/10C/10F | 10A/10B carries gene *wcrG*, whereas 10C/10F carries genes *wcrH* and *wciG* | *wcrH* encodes for GT and is responsible for side branch linkage Galf(1-6)Galp present in 10F but not in 10A; w*crG* encodes for GT and it catalyzes the linkage of Galp( 1-6) side branch in 10A (*Aanensen et al., 2007*) |
| | 10A/10B/10C/10F | 10A/10C have *wcrCα* whereas 10B/10F have *wcrCβ* | *wcrCβ* allele is described as *wcrF* and both genes encode for glucosyltransferases and are responsible for the differences observed in the linkage between galactose and ribitol-5-phosphate (*Yang et al., 2011*) |
| 11 | 11A/11B/11C/11D/11F | Genes *wcwC* and *wcjE* are present in 11A, 11D and 11F whereas gene *wcwR* is present in 11B and 11C (*Mavroidi et al., 2007*) | *wcwC*, *wcjE* and *wcwR* are acetyltransferase genes—differences in acetylation |
| | 11A/11B/11C/11D/11F | Frameshift mutation delA 130 in *gct* in 11B and 11F | Presence of Gro-1P correlates with an intact *gct* gene in types 11A and 11C; *gct* is frameshifted in types 11F and 11B, and Rib-ol is present in the CPS instead of Gro (*Mavroidi et al., 2007*) |
| | 11A/11D/11F | *wcrL pos 334*: codon AAT (Asn) in 11A; codon ACT (Ser) in 11D (Oliver et al., 2013) and codon GCT (Ala) in 11F | *wcrL* encodes for a GT—donor sugar for WcrL is GlcpNAc in types 11F, 11B, and 11C but Glcp in type 11A (*Mavroidi et al., 2007*) |
| 12 and serotypes 44 and 46 | 12A/12B/12F/44/46 | SNPs in genes *wcxD* and *wcxF* (Table S3) | Both genes encode for GTs present only in this genogroup (*Mavroidi et al., 2007*)—effect on sugar chain unknown (no structure for 12B, 44 and 46) |

**Table 3** (*continued*)

| Serogroup | Serotype | Distinguishing genetic features | Functional effect |
|---|---|---|---|
| 15 | 15A/15B/15C/15F | 15F has 4 additional genes; *glf*, *rmlB*, *rmlD* and *wcjE* (*Bentley et al., 2006*) | *glf*, *rmlB* and *rmlD* are involved in sugar biosynthesis; *wcjE* encodes for an acetyltraferase. |
| | 15A/15B/15C | *15A wchL* has 81% identity in the first 300 bps compare to the allele found in 15B/15C, whereas 15A *wzd* has 69% identity in the last 300 bps when compared to the 15B/C allele | *wchL* encodes for a GT; *wzd* is involved in translocation of mature CPS to the cell surface and thus is responsible for determining the length of the capsule polysaccharide chain (*Bentley et al., 2006*) |
| | 15B/15C | Difference in TA tandem repeat region near position 413 of *wciZ*, leading to frameshift in 15C (*Bentley et al., 2006*) | *wciZ* encodes for an O-acetyltransferase—differences in acetylation. *15B, 15B/C and 15C results can be assigned (see 'Results') |
| 16 | 16A, 16F | Mapping only | |
| 17 | 17A, 17F | Mapping only | |
| 18 | 18A/18B/18C/18F | 18F has an extra acetyltransferase gene (*wcxM*) and type 18A lacks the acetyltransferase gene *wciX* (*Mavroidi et al., 2007*) | Differences in acetylation |
| | 18B/18C | G > T 168 in *wciX* leads to early stop codon in 18B (*Mavroidi et al., 2007*) | *wciX* encodes for an acetylotranferase—difference in acetylation |
| 19 | 19A, 19F | Mapping only | |
| | 19B/19C | 19B lacks genes *wchU*, *HG264* and *glf* | *wchU* encodes for a putative GT and could be responsible for the additional glucose in the capsular polysaccharide repeat unit of 19C; *glf* encodes for a UDP-galactopyranose mutase whereas *HG264* has no functional product |
| 22 | 22A/22F | *wcwA* and *wcwC* share no similarity between 22A and 22F. | *wcwA*, encoding for a putative glycosyl-transferase and *wcwC*, encoding for a putative O-acetyltranferase—structure for 22A unknown |
| 23 | 23A/23B/23F | distinct *wzy* sequence in all serotypes | wzy encodes for a polymerase and differences in sequence should account for the different polymerization linkages (*Mavroidi et al., 2007*)—structures for 23A and 23B unknown |
| | 23A/23B/23F | *wchA* is identical in 23B and 23F but distinct in 23A. | *wchA* encodes for a glycosyl-1-phosphatase transferase (*Aanensen et al., 2007*)—structures for 23A and 23B unknown |
| 25 and serotype 38 | 25A/25F/38 | *wcyV* missing in 38 (*Mavroidi et al., 2007*) | |
| | 25A/25F/38 | *wcyDα* in serogroup 25 and *wcyDβ* in serotype 38 | *wcyV*, wcyD *and* wcyC encode for GTs (*Aanensen et al., 2007*)—no structures available for 25A, 25F or 38 |
| | 25A/25F/38 | SNPs in *wcyC* (Table S4) | |
| 28 | 28A/28F | SNPs in *wciU* (Table S5) | *wciU* encodes for a GT—no structures available |

**Table 3** (*continued*)

| Serogroup | Serotype | Distinguishing genetic features | Functional effect |
|---|---|---|---|
| 33 and serotype 37 | 33A/33F/37 | 37 carries *tts* - a transferase gene | *tts* is responsible for the polysaccharide capsule synthesis in 37 (*Waite et al., 2003*) |
| | 33A/33F | Frameshift mutation insT 433 in 33F *wcjE* gene | Loss of function of O-acetyltranferase leads to differences in acetylation (*Mavroidi et al., 2007*) |
| | 33B/33D | *wciNα* in 33B/*wciNβ*in 33C | wciNα encodes for a putative glycosyltranferase whereas wciNα encodes for a putative galactosyltransferase—consistent with differences in structure |
| | 33C | Mapping only | |
| 35 and serotype 42 | 35B, 35F | Mapping only | |
| | 35A/35C/42 | SNPs in genes *mnp1*, *wcrL* and *wzh* (Table S6) | *mnp1* encodes for a putative nucleotidyltranferase (NDP-mannitol pathway), *wcrL*, a GT and *wzh*, a protein-tyrosine phosphatase—consistent with differences in structure |
| | 35A/35C/42 | Frameshift mutation insA 248 in 35A *wcrK* (*Mavroidi et al., 2007*) | wcrK encodes for a GT—consistent with differences in structure |
| 41 | 41A/41F | Frameshift mutation delG 23 in 41A *wcrX* (*Mavroidi et al., 2007*) | *wcrX* encodes for a acetyltranferase—differences in acetylation |
| 47 | 47A, 47F | Mapping only | |

only a percentage of the reads carrying the insertion (60–70%). All 7A isolates were retested in the laboratory using the SSI sera and only the two isolates with the mixed profile were re-typed as 7A whereas the 7/9 with a wt *wcwD* were typed as 7F, suggesting that this mixed profile is indicative of 7As and can still lead to phenotypic changes. In 15Cs, differences in the length of the TA tandem repeat region of the *wciZ* gene lead to loss of function for the O-acetyltransferase in 15C and are responsible for the differences in acetylation between 15B and 15C isolates (*Bentley et al., 2006*). However, unlike the 7A/7F scenario where the two types can be easily distinguished, distinguishing 15B and 15C isolates on the bench using SSI sera in the slide agglutination method is extremely challenging. Reversible switching between the 15B and 15C serotypes was previously described in the laboratory and in natural infection (*Venkateswaran, Stanton & Austrian, 1983*; *Van Selm et al., 2003*). Thus, in practice it may not be possible to clearly distinguish between these two serotypes in the laboratory using serological methods and for the reasons above such isolates appear as 15B/C in many publications (*Laufer et al., 2010*). Therefore, it was unsurprising that the capsular typing tool had problems matching the serological results where isolates had previously been reported as either 15B or 15C prior to a change to the reporting; the previous result mostly likely due to the dominance of one type over the other in the mix. During evaluation with 30 isolates (15B, $n = 16$; 15C $n = 14$), we encountered 15B and mixed 15B/C isolates but no pure 15C isolates. A genetically pure 15C was found later during validation but a different frameshift mutation was responsible for the inactivation of *wciZ*. Overall 11/16 15B isolates were correctly predicted as 15B and 5/16 were mixed 15B/C whereas 10/14 15C isolates were predicted as 15B and 4/14 as 15B/C. Since any 15B-15C-15B/C discordances cannot be resolved in the laboratory the 15B-15B/C-15C isolates will always be reported as 15B/C.

**Table 4 Factor serum reactions for novel serogroup 7 serotype compared to other types within the group.**

|  | 7b | 7c | 7e | 7f |
|---|---|---|---|---|
| 7F | + | − | − | − |
| 7A | (+) | + | − | − |
| 7B | − | − | + | − |
| 7C | − | − | − | + |
| Novel | − | − | + | + |

## Molecular type 6E

Serotype 6E is not a serologically distinct serotype, but has been shown to be a distinct molecular subtype (*Ko, Baek & Song, 2013*). Due to the low genetic similarity between the 6E capsular locus sequence and the other serogroup 6 capsular locus sequences, PneumoCaT was unable to determine a serotype for isolates with 6E capsular locus in the first step of the capsular typing tool due to coverage <90% ($n = 12$; 10 6B and 2 6A). In order to take account of this variation, the genomic sequences of these 12 isolates were used to determine a consensus 6E capsular locus sequence and this was introduced into the capsular locus typing tool. With this in place the tool can determine whether an isolate has a 6E molecular type and using the variant profiles described above can then determine the serological type (i.e., 6E(6B)).

## Novel serogroup 7 subtype

Genogroup 7B, 7C and 40 can be distinguished using a panel of 9 SNPs (Table S3) all found within glycosyltransferase gene *wcwK*. Nucleotide differences at residues 46 and 385 are considered as more significant, as both result in distinct amino acids in the three serotypes. During evaluation using a cohort of 38 isolates (7B, $n = 15$; 7C, $n = 21$; 40, $n = 2$—Table 1 -Development Set), one isolate originally typed as serotype 7C in the laboratory, exhibited a novel codon at residue 385 which encoded for a distinct amino acid (CTT (Leu) compared to ACT (Thr) for serotype 40, TTT (Phe) for 7B and TGT (Cys) for 7C). The rest of the SNP profile matched the 7B profile. Further investigation revealed a distinct combination of factor sera reactions (Table 4) for this isolate suggesting that it might correspond to a novel serogroup 7 serotype.

## Serogroup 22

Biochemical or genetic differences between serotypes 22A and 22F have not been previously described. Initial genetic analysis revealed 99% identity over 100% of the capsular locus sequences of the two types (22A: CR931681; 22F: CR931682) suggesting a possible error in submission or serotyping. To investigate this, genomic data from the two SSI type strains (SSI-22A, SSI-22F) were assembled and using the reference sequence for 22A, the capsular locus sequences were extracted. The extracted sequences were compared to the reference sequences and the capsular locus sequence of SSI-22A was identical to the two reference sequences whereas the SSI-22F had 98% identity over 91% of the capsular locus of all three of the other sequences. Specifically, the region covering genes *wcwA* and *wcwC* shares no similarity between SSI-22F and 22A. The SSI-22F capsular locus sequence was submitted to ENA

as assembly under PHE Pathogens BioProject PRJEB14267 (Accession number LT594600.1).

## Novel molecular subtype 23B1

Serotypes 23A, 23B and 23F have distinct capsular locus sequences which allows for serotype prediction based on mapping coverage alone. A cohort of 72 isolates (23A, $n = 25$; 23B, $n = 25$; 23F, $n = 22$) was used to test the coverage-based approach. All isolates typed as 23A or 23F and 21/25 isolates typed as 23B were correctly predicted; however, $4 \times 23$B isolates gave a failed result in the capsular typing tool with highest coverage of approximately 70%. These isolates were re-tested in the laboratory using SSI sera and the previous results were confirmed. In order to investigate this further, more 23B isolates were sequenced and analysed using the capsular typing tool. In total, 24/46 23B isolates failed the pipeline with 70% coverage. Following further investigation, these isolates were identified as a novel molecular subtype (23B1) with ∼70% homology to the 23B capsular locus. The 23B1 capsular locus sequence was introduced into PneumoCaT assuring 100% concordance with serological type.

## Serogroup 24

Genetic differences between serotypes in serogroup 24 have previously been described (*Mavroidi et al., 2007*) and were also confirmed following genetic analysis of the reference sequences (Accession numbers CR931686, CR931687, CR931688 for 24A, B and F, respectively). Serotype 24A lacks *wzy*, a polymerase gene and *rbsF*, a putative ribofuranose biosynthetic gene. In addition, the gene sequence of the flippase gene *wzx* is unique to 24A as indicated by using a blast nucleotide query against the NCBI database. This could only be confirmed by the SSI type strain (SSI-24A) since no other 24A isolates were available in the PHE archive. According to *Mavroidi et al. (2007)* serotypes 24B and 24F are distinguished by loss of function of genes *abp1* and *rmlC* in 24B. This was confirmed when comparing the reference sequences. However, following evaluation with a cohort of 25 isolates (6, 24B; 19, 24F) the frameshift in *rmlC* was not seen in any of the 24B isolates, including the SSI-24B type strain whereas the frameshift in *abp1* was only seen in SSI-24B strain, but no other 24B strains. This suggested that either all 24B isolates are actually 24F or that the reference sequences are not representative of the circulating strains; the serological typing was repeated for all isolates and original serotypes were confirmed, therefore we surmised that the latter is more likely. Genomic SNP analysis was used to investigate this further by mapping to either (a) the 24F capsular locus sequence (CR931688) or (b) the whole genome of the non-capsular *S. pneumoniae* R6 strain (NC_003098). Subsequent variant calling and filtering produced 145 and 19,123 high quality SNPs, respectively. Evolutionary history was inferred using the Maximum-likelihood method and phylogenetic trees were drawn (Fig. S1). No serotype specific clustering was observed in either of the trees suggesting that if there is a molecular basis for the observed serological differences, it occurs using a mechanism that is not vertically inherited e.g rapid strand-slippage variation in a microsatellite within a promoter region. Further investigation is underway but in the meantime, PneumoCaT can predict isolates to serogroup level and serological analysis can distinguish further to serotype

level. Although genetic differences have been described above for differentiating 24A from 24B/24F, they are not currently implemented in PneumoCaT due to incomplete evaluation.

## Serogroup 32

Structures for the capsular polysaccharide repeat unit of both serotype 32A and 32F are available and differences based on the structure have previously been described (*Bentley et al., 2006*; *Mavroidi et al., 2007*). However, when the two reference capsular locus sequences (32A: CR931696; 32F: CR931697) were aligned, 99% identity was observed over 99% of the length of the sequence. The only difference was a 5 bp gap at the intergenic region between *wcrN* and the *HG272/3* pseudogene observed in the serotype 32 capsular locus. Gene-based analysis returned no nucleotide differences within coding regions. To investigate whether there was a submission error, as described for 22F, the capsular locus sequences (lacking the *tnp* regions at the beginning of the capsular locus) of the 32A and 32F SSI type strains were extracted from the assembled genomes of SSI-32A and SSI-32F strains and compared to the reference strains. Both type sequences showed 100% identity with the respective reference sequences over the length of the alignment. The serogroup 32 evaluation cohort only includes 5 isolates (32A, $n = 2$; 32F, $n = 3$—Table 1; Development Set) and following analysis with the capsular typing tool all isolates were assigned to serogroup 32. Further investigation into the BAM files revealed that the 5 bp gap was seen only in the SSI-32F type strain. More serogroup 32 isolates need to be analysed before any conclusions can be reached, but no serogroup 32 isolates have been observed in the UK since 2007. At present, the predicted serogroup 32 isolates can only be differentiated into serotypes using serological analyses.

## Validation of the capsular typing tool

Following development, the PneumoCaT tool was evaluated using two panels: (a) 2046 UK isolates retrieved from PHE archive covering 72/92 serotypes, including all serotypes contained in commercial vaccines (Table 5; excluding non-typeables) and (b) genomic data from 2964 non-UK isolates from three distinct, publicly available datasets (Thailand; $n = 2531$—65 serotypes, USA; $n = 181$—serotype information not available- and Iceland; $n = 252$—serotypes 6A, 6B, 6C and 6E).

Both panels were used to evaluate the typability of PneumoCaT based on the proportion of isolates that were assigned a type. In the UK cohort, serotype was assigned initially to 98.2% of the typeable isolates (2010/2046). Of the 36 failed isolates, 19 were called as mixed and following retesting using a culture from a single colony, 18/19 were resolved, raising the typability to 99.1%. In the non-UK cohort, a serotype was assigned to 99% of the isolates (2934/2964) and 8/20 failed ones were called as mixed, suggesting that if retesting was possible these could be resolved.

Concordance with currently used methodology (slide agglutination using SSI sera) was evaluated using the UK cohort for which further laboratory testing was possible. Concordance was estimated on the 2028 typeable isolates for which a serotype was called with both methods (2010 + 18 mixed resolved). Overall, 92.4% concordance was observed between the predicted serotypes and the serologically-derived serotypes ($n = 1873$). Discordance was observed in 155 isolates (Table 6) and in most cases (77%, $n = 119$), the
**Table 5  Detail breakdown of the concordance analysis for the UK validation panel (n = 2,065; 2,046 typeable and 19 non-typeables).**

| Serotype | Total | Initial | | | After retesting | | |
|---|---|---|---|---|---|---|---|
| | | Concordant | Discordant | Failed WGS | Concordant | Discordant | Non-typeable |
| 1 | 41 | 40 | 1 | | 40 | | |
| 2 | 9 | 8 | 1 | | 8 | | |
| 3 | 44 | 44 | | | 46 | | |
| 4 | 43 | 43 | | | 44 | | |
| 5 | 41 | 40 | 1 | | 40 | | |
| 8 | 70 | 70 | | | 72 | | |
| 13 | 20 | 20 | | | 20 | | |
| 14 | 43 | 40 | 3 | | 40 | 1 | |
| 20 | 42 | 39 | 3 | | 40 | | |
| 21 | 21 | 21 | | | 21 | | |
| 27 | 23 | 23 | | | 24 | | |
| 29 | 21 | 9 | 11 | 1 | 9 | 1 | 1 |
| 31 | 22 | 22 | | | 23 | | |
| 34 | 22 | 22 | | | 24 | | |
| 36 | 5 | 3 | 1 | 1 | 3 | | |
| 37 | 22 | 21 | 1 | | 22 | | |
| 38 | 23 | 19 | 1 | 3 | 19 | | 1 |
| 39 | 2 | 2 | | | 2 | | |
| 40 | | | | | 1 | | |
| 42 | | | | | | | |
| 43 | | | | | | | |
| 44 | | | | | | | |
| 45 | | | | | | | |
| 46 | | | | | | | |
| 48 | 5 | 5 | | | 5 | | |
| 06A | 41 | 41 | | | 42 | | |
| 06B | 43 | 37 | 6 | | 37 | | |
| 06C | 22 | 21 | 1 | | 24 | | |
| 06D | 2 | 1 | 1 | | 2 | | |
| 07A | 5 | 1 | 4 | | 1 | | |
| 07B | 4 | 3 | 1 | | 3 | | |
| 07C | 24 | 22 | 2 | | 22 | | |
| 07F | 40 | 38 | 2 | | 43 | | |
| 09A | 6 | 1 | 5 | | 1 | | 1 |
| 09L | 2 | 1 | 1 | | 1 | | |
| 09N | 44 | 42 | 2 | | 43 | | |
| 09V | 45 | 45 | | | 49 | | |
| 10A | 44 | 41 | 3 | | 42 | 1 | |
| 10B | 7 | 6 | 1 | | 6 | | |
| 10C | | | | | | | |

**Table 5** (*continued*)

| Serotype | Total | Initial | | | After retesting | | |
|---|---|---|---|---|---|---|---|
| | | Concordant | Discordant | Failed WGS | Concordant | Discordant | Non-typeable |
| 10F | 22 | 19 | 1 | 2 | 20 | | |
| 11A | 44 | 40 | 2 | 2 | 46 | | |
| 11B | 2 | 2 | | | 2 | | |
| 11C | 4 | | 4 | | | | |
| 11D | | | | | | | |
| 11F | | | | | | | |
| 12A | 2 | | 1 | 1 | | | |
| 12B | 23 | 5 | 18 | | 9 | | |
| 12F | 44 | 38 | 6 | | 58 | | 2 |
| 15A | 196 | 191 | 4 | 1 | 200 | | |
| 15B[a] | 41 | 40 | | 1 | 42 | | |
| 15B/C[a] | 8 | 8 | | | 26 | | |
| 15C[a] | 24 | 16 | 6 | 2 | 1 | | |
| 15F | 2 | | 2 | | | | |
| 16A | | | | | | | |
| 16F | 21 | 21 | | | 25 | | |
| 17A | | | | | | | |
| 17F | 41 | 41 | | | 42 | | |
| 18A | 11 | 11 | | | 11 | | |
| 18B | 9 | 8 | 1 | | 8 | | |
| 18C | 41 | 38 | 3 | | 39 | 3 | |
| 18F | 5 | 3 | 2 | | 3 | | |
| 19A | 249 | 245 | 3 | 1 | 246 | | |
| 19B | 1 | 1 | | | 1 | | |
| 19C | 1 | | 1 | | | | |
| 19F | 41 | 41 | | | 42 | | |
| 22A | 2 | 2 | | | 2 | | |
| 22F | 43 | 43 | | | 47 | | |
| 23A | 23 | 22 | 1 | | 25 | | |
| 23B | 79 | 78 | 1 | | 90 | | |
| 23F | 40 | 27 | 11 | 2 | 27 | | |
| 24A | 2 | | 2 | | | | |
| Serogroup 24 | 31 | 30 | 1 | | 34 | | |
| 25A | 1 | | 1 | | | | 1 |
| 25F | 1 | 1 | | | 1 | | |
| 28A | 19 | 19 | | | 19 | | |
| 28F | 1 | | 1 | | | | |
| Serogroup 32 | | | | | | | |
| 33A | 5 | | 5 | | | | 2 |
| 33B | 1 | 1 | | | 1 | | |
| 33C | 1 | | | 1 | | | |
| 33D | | | | | | | |
| 33F | 43 | 43 | | | 48 | | |
| 35A | 24 | 4 | 20 | | 4 | | 1 |

**Table 5** (*continued*)

| Serotype | Total | Initial | | | After retesting | | |
|---|---|---|---|---|---|---|---|
| | | Concordant | Discordant | Failed WGS | Concordant | Discordant | Non-typeable |
| 35B | 23 | 23 | | | 52 | | |
| 35C | 4 | | 4 | | | 1 | 1 |
| 35F | 22 | 21 | 1 | | 22 | | |
| 41A | 1 | | 1 | | | 1 | |
| 41F | | | | | | | |
| 47A | | | | | | | |
| 47F | | | | | | | |
| NOVEL 9[b] | | | | | 1 | | |
| Total | 2,046 | 1,873 | 155 | 18 | 2,013 | 8 | 10 |
| NT | 19 | 13 | 6 | | 13 | 3 | |
| Grand Total | 2,065 | 1,886 | 161 | 18 | 2,026 | 11 | 10 |

Notes.
[a]If not possible to resolve discordance using serology, a 15B/C serotype is reported.
[b]Novel 9 serotype by WGS and SSI sera.

discordance involved serotypes using the same factor sera set for serotype allocation (i.e., 12B/12F, 7A/7F discordances) or serotypes using the same pool sera (i.e., serotype 31 and 7F both react with pool C serum) (*Statens Serum Institut, 2013*) ($n = 22$). Repeating the serological serotyping for these two sets resolved the initial discordance in 88.7% of the cases (125/141). This suggests that discordance in these cases may be attributed to difficulty in reading slide agglutination reactions during the serological analysis with the SSI sera (i.e., auto-agglutination with all group 12 factor sera in the case of 12B/12F typing). In one case, a novel serogroup 9 serotype was predicted based on both the SNP-pattern (Table S4) and sera reaction pattern that differed from the expected 9L pattern, showing a weak reaction with factor sera 9 g. In the remaining 14/155 cases, there are no shared sera between the serotypes predicted by WGS and the serologically-derived serotypes. These isolates were retested with both methods and all but one (unable to serotype due to autoagglutination) were concordant suggesting a possible laboratory sample translocation during archiving and/or retrieval of the isolate. Overall, following retesting, we observed a final concordance of 99.3% (2013/2028), with only 18 discordant. In some of these cases, serotyping could not resolve the discordance because auto-agglutination leads to inconclusive results ($n = 10$), whereas other cases ($n = 8$) exhibit a persistent discordance even after retesting with both methods (Table S9). In these cases, the capsular operon sequence was investigated further using an assembly and blast approach to query for both capsular operons. In all cases, the capsular operon sequence detected did not match the serologically derived serotype. However, it must be noted that for each of these serotypes, serotype was accurately predicted for the majority of cases and these discordant cases are outliers (Table 5).

The serogroup 6 isolates from the Icelandic dataset ($n = 252$) were originally used to demonstrate the distribution of the new 6E type and molecular typing was used to characterized all isolates (*Van Tonder et al., 2015*). Although these isolates were not available for further investigation, they were the only complete dataset with confirmed 6E isolates

**Table 6** Investigation of discordant isolates in the validation set.

| Isolate | Slide agglutination | | WGS | |
|---|---|---|---|---|
| | Initial | Repeat | Initial | Repeat |
| PHESPV0001 | 18F | 3 | 3 | |
| PHESPV0002 | **09L** | Novel 9 pattern | Novel 9 | Novel 9 |
| PHESPV0012 | **09A** | 09V | 09V | |
| PHESPV0013 | **09A** | 09V | 09V | |
| PHESPV0016 | **12B** | 12F | 12F | 12F |
| PHESPV0024 | **35C** | 35B | 35B | |
| PHESPV0025 | **07A** | 07F | 07F | |
| PHESPV0027 | **09A** | 09V | 09V | |
| PHESPV0029 | **18B** | 18C | 18C | |
| PHESPV0070 | **06B** | 06D | 06D | |
| PHESPV0071 | NT | 27 | 27 | |
| PHESPV0101 | **09N** | 09N | 23B | 09N |
| PHESPV0112 | 06B | 17F | 17F | |
| PHESPV0121 | 09N | 8 | 8 | |
| PHESPV0128 | 1 | 15A | 15A | |
| PHESPV0174 | **09A** | 09V | 09V | |
| PHESPV0189 | 2 | 11A | 11A | |
| PHESPV0194 | 29 | 22F | 22F | |
| PHESPV0197 | **29** | 35B | 35B | |
| PHESPV0200 | 29 | 34 | 34 | |
| PHESPV0204 | 36 | 22F | 22F | |
| PHESPV0209 | **23F** | 23B | 23B | |
| PHESPV0211 | **29** | 35B | 35B | |
| PHESPV0235 | NT | 35B | 35B | |
| PHESPV0303 | **07A** | 07F | 07F | |
| PHESPV0322 | **15A** | 15B/C | 15B | |
| PHESPV0348 | **24A** | 24F | Serogroup 24 | |
| PHESPV0349 | **35A** | 35B | 35B | |
| PHESPV0380 | **24A** | 24F | Serogroup 24 | |
| PHESPV0438 | **15A** | 15B/C | 15B | |
| PHESPV0448 | **15F** | 15B | 15B | |
| PHESPV0458 | **23F** | 23B | 23B | |
| PHESPV0489 | **35A** | 35B | 35B | |
| PHESPV0518 | **35A** | 35B | 35B | |
| PHESPV0525 | **35A** | 35B | 35B | |
| PHESPV0526 | **35A** | 35B | 35B | |
| PHESPV0537 | **15A** | 15B/C | 15B/C | |
| PHESPV0555 | **06B** | 06C | 06C | |
| PHESPV0556 | **23F** | 23B | 23B | |
| PHESPV0557 | **35A** | 35B | 35B | |

Kapatai et al. (2016), *PeerJ*, DOI 10.7717/peerj.2477

**Table 6** (*continued*)

| Isolate | Slide agglutination | | WGS | |
| --- | --- | --- | --- | --- |
| | Initial | Repeat | Initial | Repeat |
| PHESPV0558 | **11C** | 11A | 11A | |
| PHESPV0562 | **15F** | 15B | 15B | |
| PHESPV0563 | **33A** | 33F | 33F | |
| PHESPV0566 | **23F** | 23B | 23B1 | |
| PHESPV0582 | **19A** | 19F | 19F | |
| PHESPV0586 | **06B** | 06C | 06C | |
| PHESPV0591 | **35A** | 35B | 35B | |
| PHESPV0605 | **35A** | 35B | 35B | |
| PHESPV0608 | 20 | 11A | 11A | |
| PHESPV0613 | **35A** | 35B | 35B | |
| PHESPV0616 | 06B | 33F | 33F | |
| PHESPV0624 | **35A** | 35B | 35B | |
| PHESPV0654 | **35A** | 35B | 35B | |
| PHESPV0663 | **35A** | 35B | 35B | |
| PHESPV0667 | **12B** | 12F | 12F | |
| PHESPV0669 | **35A** | 35B | 35B | |
| PHESPV0678 | **23F** | 23B | 23B | |
| PHESPV0681 | **12B** | 12F | 12F | |
| PHESPV0691 | NT | 37 | 37 | |
| PHESPV0698 | 24F | 07F | 07F | |
| PHESPV0700 | 07F | 20 | 20 | |
| PHESPV0744 | 10A | 16F | 16F | |
| PHESPV0753 | **35A** | 35B | 35B | |
| PHESPV0761 | **12B** | 12F | 12F | 12F |
| PHESPV0773 | **35A** | 35B | 35B | |
| PHESPV0779 | **12B** | 12F | 12F | |
| PHESPV0780 | **12B** | 12F | 12F | |
| PHESPV0794 | **12B** | 12F | 12F | |
| PHESPV0796 | **12B** | 12F | 12F | 12F |
| PHESPV0797 | **12B** | 12F | 12F | |
| PHESPV0804 | **35A** | 35B | 35B | |
| PHESPV0805 | **33A** | 33F | 33F | |
| PHESPV0812 | **10B** | 10F | 10F | |
| PHESPV0820 | **12A** | 12F | 12F | |
| PHESPV0836 | **23F** | 23B | 23B | |
| PHESPV0839 | **11C** | 11A | 11A | |
| PHESPV0840 | **33A** | 33F | 33F | |
| PHESPV0843 | **35A** | 35B | 35B | |
| PHESPV0845 | **23F** | 23A | 23A | |
| PHESPV0854 | **06B** | 06A | 06A | |

**Table 6** (*continued*)

| Isolate | Slide agglutination | | WGS | |
|---------|-------------|-------------|-------------|-------------|
| | **Initial** | **Repeat** | **Initial** | **Repeat** |
| PHESPV0861 | **35A** | 35B | 35B | |
| PHESPV0862 | 18F | 23B | 23B1 | |
| PHESPV0888 | **11C** | 11A | 11A | |
| PHESPV0891 | **11C** | 11A | 11A | |
| PHESPV0894 | 37 | 16F | 16F | |
| PHESPV0898 | 19A | 3 | 3 | |
| PHESPV0912 | 14 | 15B | 15B | |
| PHESPV1001 | **29** | 35B | 35B | |
| PHESPV1005 | 14 | 12F | 12F | 12F |
| PHESPV1014 | **35A** | 35B | 35B | |
| PHESPV1017 | 23F | 35F | 35F | |
| PHESPV1018 | 14 | 23A | 23A | |
| PHESPV1020 | **23F** | 23B | 23B1 | |
| PHESPV1084 | 20 | 22F | 22F | |
| PHESPV1140 | **12F** | 12B | 12B | 12B |
| PHESPV1168 | 15A | 15A | 19F | 15A |
| PHESPV1178 | 06C | 23B | 23B1 | |
| PHESPV1200 | 07B | 24F | Serogroup 24 | |
| PHESPV1208 | **12F** | 12B | 12B | 12B |
| PHESPV1244 | 23A | 15A | 15A | |
| PHESPV1283 | **12F** | 12B | 12B | |
| PHESPV1390 | 07F | 31 | 31 | |
| PHESPV1401 | 10A | 22F | 22F | |
| PHESPV1406 | **12F** | 12B | 12B | |
| PHESPV1413 | 19A | 8 | 8 | |
| PHESPV1418 | **06D** | 06C | 06C | |
| PHESPV1648 | **23F** | 23B | 23B | |
| PHESPV1650 | **23F** | 23B | 23B1 | |
| PHESPV1652 | **07A** | 07F | 07F | |
| PHESPV1654 | 35F | 34 | 34 | |
| PHESPV1668 | 11A | 16F | 16F | |
| PHESPV1716 | **07A** | 07F | 07F | |
| PHESPV1734 | 11A | 16F | 16F | |
| PHESPV1790 | 20 | 24F | Serogroup 24 | |
| PHESPV1809 | 28F | 23B | 23B | |
| PHESPV1845 | **15C** | 15A | 15A | |
| PHESPV1847 | **15C** | 15A | 15A | |
| PHESPV1848 | **15C** | 15A | 15A | |
| PHESPV1864 | 5 | 4 | 4 | |
| PHESPV1868 | **15C** | 15A | 15A | |
| PHESPV1869 | **15C** | 15A | 15A | |
| PHESPV1874 | **15C** | 15A | 15A | |
| PHESPV1883 | **29** | 35B | 35B | |

**Table 6** (*continued*)

| Isolate | Slide agglutination | | WGS | |
|---------|---------|---------|---------|---------|
| | Initial | Repeat | Initial | Repeat |
| PHESPV1919 | **29** | 35B | 35B | |
| PHESPV1935 | **29** | 35B | 35B | |
| PHESPV1941 | **10F** | 10A | 10A | |
| PHESPV1951 | **35C** | 35B | 35B | |
| PHESPV2005 | **12B** | 12F | 12F | |
| PHESPV2007 | **12B** | 12F | 12F | |
| PHESPV2017 | **23B** | 23A | 23A | |
| PHESPV2018 | **19C** | 19A | 19A | |
| PHESPV2028 | **12B** | 12F | 12F | |
| PHESPV2032 | **29** | 35B | 35B | |
| PHESPV2035 | **12B** | 12F | 12F | |
| PHESPV2039 | **12B** | 12F | 12F | |
| PHESPV2042 | **12B** | 12F | 12F | |
| PHESPV2043 | 07C | 33F | 33F | |
| PHESPV2049 | **12B** | 12F | 12F | |
| PHESPV2052 | **12B** | 12F | 12F | |
| PHESPV2062 | **12B** | 12F | 12F | |
| PHESPV2063 | **07C** | 40 | 40 | |

(molecular confirmation since serological confirmation is not possible), so they were used to demonstrate the ability of PneumoCaT to differentiate 6E from the other serogroup 6 types. Following analysis, 100% typability and concordance was observed ($n = 252$).

Reproducibility of the WGS method was investigated using a subset of 292 UK isolates covering 82/94 serotypes that were randomly selected for inclusion in the reproducibility study. For this study each isolate was cultured twice and DNA extracted and sequenced at different times. Concordance was seen in all cases.

## Non typeable isolates

Virulence of *S. pneumoniae* is classically associated with the capsule, however some pathogenic strains, often associated with non-invasive sites (i.e., respiratory isolates), are serologically non-typeable (NT) (*Park et al., 2012*). Nineteen NT strains were selected from the PHE archive and analysed using WGS and PneumoCaT to predict serotype. In 13/19 cases PneumoCaT failed to predict serotype with a highest coverage <50% (Fig. 4 and Table 5). This is consistent with lack of a capsular operon and these isolates would be considered non-typeable. In addition, three isolates that originally had been typed in the lab (19A and 2 × 38) but failed with <50% coverage during PneumoCaT analysis were retested and found non-typeable in the lab, suggesting that this cut off could be used an indicative of non-typeable isolates following further validation. In the remaining 6/19 cases a serotype was predicted. In three of the cases, retesting with SSI sera gave the same serotype as predicted by PneumoCaT. In three other cases, repeat serotyping by slide agglutination

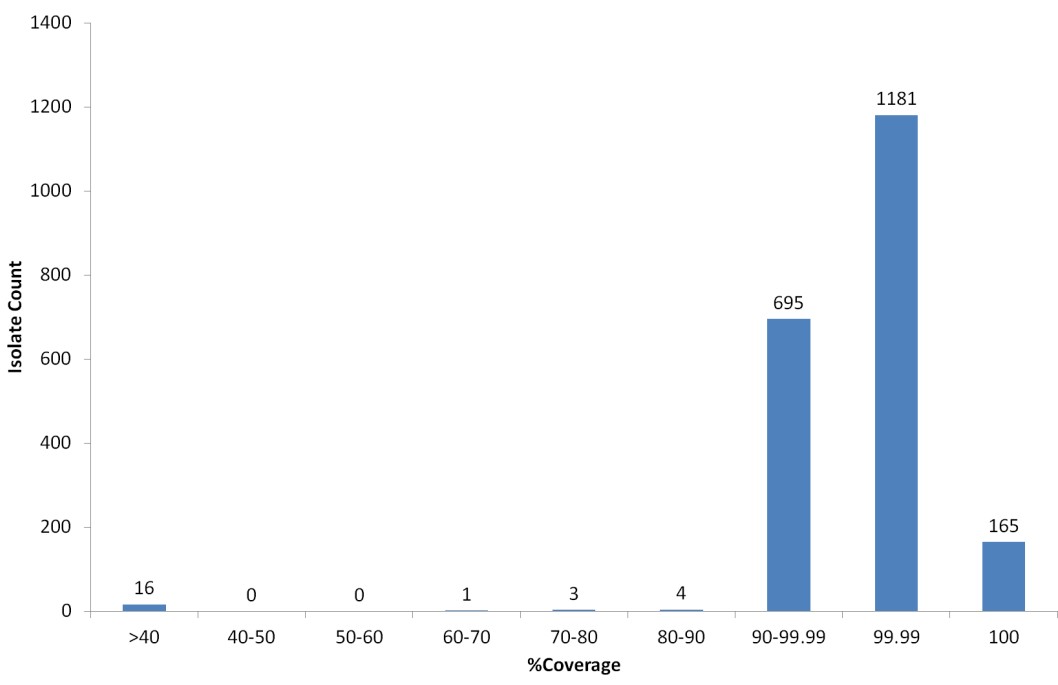

**Figure 4** Percent coverage distribution of top serotype hit in PneumoCaT stage 1 for all *S. pneumoniae* isolates (*n* = 2065) in the study.

consistently failed to give an acceptable result. In all of these cases, the colonies had an abnormal morphology (i.e., rough).

## DISCUSSION

Molecular capsular typing methods, such as multiplex PCR, have been readily used in reference laboratories across the globe in place of the gold standard Quellung method ("*WHO, 2011*—Laboratory methods for the diagnosis of meningitis caused by Neisseria meningitidis, *Streptococcus pneumoniae*, and Haemophilus influenzae"; *European Centre for Diseases Prevention and Control (ECDC), 2013*). The lower cost and increased turnaround times associated with such methods have made them the preferred serogrouping/typing method for a number of laboratories. Many conventional and real-time PCR schemes cover the vaccine-related and most commonly observed serotypes (*Jauneikaite et al., 2015*). However, these methods do not cover all serotypes and many types cannot be distinguished from others with genetically closely related capsular operons. In this study, we present serotype-specific variant profiles that can be used to distinguish 87/92 serologically-distinct serotypes and 2 molecular subtypes, and PneumoCaT, a bioinformatics tool that uses the CTV (Capsular Typing Variant) database to predict capsular type from WGS data.

A gene-based approach was used to identify variants that allow differentiation between closely-related serotypes. During its development stage, a test cohort of 871 clinical isolates, including the SSI-type strains, was used to evaluate the relevance of these variants in the population and establish the CTV database. In many cases, these sequence variants were

previously published (e.g., serogroup 6) (*Song, Baek & Ko, 2011*), whereas in other cases structural differences have been observed but no molecular information was provided (e.g., serogroup 9). One of the main advantages of this approach is the ability to detect molecular subtypes (and potential new phenotypes) even if no serological differences are observed using standard typing sera. For example, isolates from serogroups 6 and 23 initially failed the development pipeline due to low capsular locus coverage. Further investigation into the reason for the failures was able to identify serotype 6E, a previously described molecular type (*Ko, Baek & Song, 2013*) and a novel 23B molecular subtype. Another example is serotype 19A, where two subgroups were identified based on the mapping coverage of the 19A capsular locus. In this case, all isolates were still predicted as 19A by PneumoCaT, since more than 90% coverage was observed for both subgroups, but gene profile analysis revealed variability in the presence of cps19aO gene and further functional and epidemiological analysis is currently underway to investigate whether this has any phenotypic relevance. In terms of surveillance, the presence of these new molecular subtypes could indicate introduction of new clones that may influence new pneumococcal vaccine development. In some rare cases, a novel molecular type can also have corresponding serological differences; for example, during the test phase of the variant profiles for genogroup 7B-7C-40, an isolate exhibited a distinct codon at position 385 and a distinct factor sera reaction pattern (Table 4) upon retesting in our laboratory. Following further investigation by SSI this isolate has now been confirmed as a novel serotype (authors' unpublished data). In addition, during the validation phase of this study, SNP analysis revealed a novel serogroup 9 type with a distinct sera reaction pattern, but this has not been confirmed.

Following the development stage, PneumoCaT was evaluated for typability, accuracy (concordance with serological results) and reproducibility. Typability was evaluated as the percentage of isolates for which a serotype was assigned by PneumoCaT, using two panels of isolates (UK, $n = 2,046$; non-UK, $n = 2,964$), that combined, cover 82/94 capsular types, including clones that cover 4 different countries (UK, USA, Iceland and Thailand). Using both panels, a combined typability of 99% (99.1% for UK and 99% for non-UK) was achieved, demonstrating that PneumoCaT can be used to analyse isolates from diverse geographical lineages. The panel of 2,028 typeable UK isolates (encompassing 72 serotypes) was then used to evaluate the accuracy of the tool. When discordances between phenotypic serotyping results and PneumoCaT serotype were investigated, the majority (119/155) were shown to be due to problems with determining serotype within serogroup, when using the factor sera in the slide agglutination method. Particular serogroups such as 12, 33 and 35 proved especially problematic (Table 5). This could be due to the way the results are interpreted for the factor sera reactions when used in slide agglutination. For example, in 24/69 Serogroup 12 cases a 12B/12F discordance was observed; 18 serotype 12B isolates were assigned 12F capsular type by PneumoCaT whereas 6 12F isolates were assigned 12B. Based on the SSI typing scheme a reaction with all three serogroup 12 factor sera (12b, 12c, 12e) is required to identify 12B, whereas a reaction with factor serum 12b only is indicative of 12F isolates. Unfortunately, some factor sera agglutination reactions can be weaker than others. Therefore, a weak reaction in all factor sera could be mistaken for auto-agglutination or *vice versa*, leading to incorrect scoring. In most discordant cases,

repeating the phenotypic testing resolved the discordance. In some cases ($n = 10$) a weak reaction or auto-agglutination made interpretation of the serotyping result difficult, thus the serotype could not be confidently called using the slide agglutination method with the standard SSI sera. The standard factor sera obtained from Statens Serum Institut are optimised for use in the Quellung reaction rather than slide agglutination and therefore, it is highly likely that if the serotyping was performed using the gold-standard Quellung method by technicians with the necessary experience, this could resolve the discrepancies in many cases and increase the accuracy of the serotyping. The use of latex absorbed sera specifically designed for agglutination reactions is also expected to improve the accuracy of the serotyping results for the more difficult to distinguish serotypes. Late in this study, a limited number of latex-absorbed subtyping sera were kindly provided by SSI for use in agglutination reactions, and were used to re-serotype all discordant serogroup 12 isolates. The latex sera was able to resolve 22/24 discordances and in 2/24 cases a serotype could not be confidently called due to autoagglutination.

In addition, some of the observed original discordances clearly came from simple record keeping errors due to the nomenclature of the factor sera being very similar to the nomenclature for the serotypes themselves, such as serotype 15A which is reacts with 15c factor sera, which accounted for several discordant cases where 15A isolates had been originally recorded as 15C due to 15c factor positive result. Similar discordances were also observed in serogroup 6 where 6A reacts with 6b factor sera, 6B with 6c and 6C with 6d factor sera resulting in mis-reporting. This kind of error could be avoided by use of automated reporting mechanisms which take electronically derived results (such as the PneumoCaT output) and enter them directly into laboratory information systems, avoiding manual transcription.

Where occasional discordance persisted even after retesting, it involved pairs of serotypes within the same serogroup or with known cross-reactions (Table S9). In these persistent cases, further molecular investigation is underway to determine whether any regions outside of the capsular operon might be involved in this discordance between phenotype and serotype. However, a repeated serotyping error with these particular isolates cannot be excluded since other examples of the same serotypes have concordance between WGS and serology. For example, one of the isolates is serologically 29 where PneumoCaT predicted 35B. The two capsular operon sequences share approximately 70% of the capsular locus sequence with 87% similarity and can be distinguished serologically based on the reaction of a single factor sera (i.e., 35a gives positive reaction if 35B and negative if 29, and there are known cross-reactions between 29 and some of the factor sera for group 35). The presence of a complete serotype 35B capsular operon in this discrepant isolate was confirmed by assembly as well as mapping and following analysis of the capsular locus sequence no differences were found between this isolate and other 35Bs, including two that were originally mistyped as 29. At this point, further analysis for external factors is underway.

Three of the persistent discordances involved serotype 18C; 2 assigned 18B and one 18A serotype by PneumoCaT. Following retesting by both WGS and sera; 2 persisted with 18C/18B discordance, whereas the third isolate was re-serotyped as 18B and retyped as 18A by WGS. The 18C/B discordance is not unexpected as both serotypes react with

18e factor sera whereas 18c sera reacts only with 18C serotypes, suggesting that a weak reaction or autoagglutination could lead to false prediction. However, 18B/A discordance is unexpected as the serotypes react with two different factor sera (18d for 18A and 18e for 18B). Interestingly, the molecular difference distinguishing 18A and 18B is the presence of a non-functional *wciX* gene in 18B; this difference should have no impact on phenotype, but has successfully differentiated 19 18A/B isolates (18A, $n = 11$ and 18B, $n = 8$). We further examined the capsular sequence to determine additional differences that could impact the phenotype. Specifically, we looked into *glf* gene, which based on previous studies (*Bentley et al., 2006*) was present as a pseudogene in 18B, 18C, and 18F. However, when these gene sequences were translated it was evident that a large part of the UDP-galactopyranase mutase domain was still encoded; specifically within the translated *glf* sequence in 18B and 18C a 167 amino acid ORF was present and a 204 amino acid ORF in 18F. Both ORFs matched to published UDP-pyranase mutases when compared to the NCBI protein database using BLAST. The genomic reads of the 19 18A/B and the discordant 18B/A isolate were mapped to the 502 nucleotide coding region of 18B *glf* and as expected *glf* was present in all 18Bs and none of the 18As. However, *glf* was also found in the 18B/A isolate which also lacks *wciX*, a genetic marker for 18A. This finding can potentially explain the discordance between phenotype and genotype and incorporation of the *glf* marker in CTV will prevent any future 18A/B discordances and instead flag them for further investigation.

Another of the discordant isolates involved a rare serotype (41A/41F) and was difficult to resolve due to the lack of representative isolates of serogroup 41. If implemented into the routine laboratory, isolates of these rare serotypes would be subjected to traditional serotyping in addition to WGS analysis until the pipeline could be further developed or confidence in the results was assured.

The high typability, concordance and reproducibility rates (99.1, 99.2 and 100%, respectively) observed with PneumoCaT suggest the method is highly robust and reliable, with less subjectivity than traditional serotyping and full traceability of results. The sensitivity of the method for serotype mixtures is greater than that of the standard method which means that careful preparation and handling of the isolates for DNA extraction and WGS is necessary to avoid cross-contamination, but also means that potentially genuinely mixed serotype cultures can be recognised. For example, an isolate was initially typed as 19A and predicted as mixed 19A/3 with WGS when retrieved from the archives for testing. It was found to have colonies with two different morphologies; one very mucoid and growing over a smaller non-mucoid colony form. The mucoid colony was sampled and following retesting serotype 3 was assigned using both traditional serotyping and PneumoCaT. Unfortunately, the smaller colonies could not be sampled in a pure form but were presumably the original serotype 19A, given the PneumoCaT result.

A further advantage of the PneumoCaT method is the extra information obtained in terms of the percentage mapping to the capsular reference and SNPs identified, which could lead to the discovery of novel serotypes. Trends in the mapping percentage (6E and 23B1) and differences in SNP patterns for those entering the CTV detection pathway could be recorded (novel serogroup 7 and serogroup 9 subtypes). These novel types may be missed by standard serotyping methods due to the way the typing sera are cross-absorbed

to avoid reactions with known types; this, however, does not avoid cross-reactions with novel types e.g., 6C which initially cross reacted with 6A serotyping factor sera prior to production of specific factor sera for this type (*Lambertsen & Kerrn, 2010*).

A small number of isolates ($n = 11$) failed during the first stage (coverage-based) of the capsular typing pipeline and, upon further examination, repeat serotyping was also unable to confidently call a type in five cases suggesting that they might be true acapsular, non-typeable isolates. In 2/5 cases, % coverage was between 75–90% suggesting that a capsular operon was present but not expressed. The remaining six isolates fall just short of the 90% coverage threshold (Fig. 4), but the top hit corresponded to the expected serotype/serogroup suggesting that a capsule could still be expressed (Table S10).

Non-typeable isolates, lacking a functional capsular operon, also get a 'Failed' tag by PneumoCaT but can easily be distinguished, based on coverage levels, from unusual isolates where serotype prediction failed. Based on our analysis of 19 isolates, serologically non-typeable isolates have a highest coverage value of less than 50% whereas for unusual isolates whereas some reactions with serotyping sera are seen when the coverage falls within 60–90%. With further data to confirm this observation, a 'Non-typeable' flag could be introduced to PneumoCaT (Fig. 4). During our analysis of these non-typeable isolates, in 32% of the cases a serotype was predicted by PneumoCaT, indicating a functional operon was present. Upon retesting, half of these gave the serotype predicted and half were still non-typeable, indicating possible external regulatory elements that inhibit expression of the capsule in these organisms.

In summary, PneumoCaT is a robust, accurate, sensitive and expandable tool that could revolutionise pneumococcal reference microbiology. This tool has the sensitivity to enable the recognition of mixed serotypes or new subtypes that could be masked by the use of other methods. The flexibility to introduce new types to the CTV database with relative ease means the system can evolve to suit future challenges. The CTV database itself has the potential to inform the further development of other molecular methods, for example PCR or microarray analysis for determination of capsular type and it could be particularly useful for developing non-culture typing schemes. PneumoCaT will enable more detailed surveillance of serotype drift and could be used to target further phenotypic analysis of potential new serotypes. The automated nature of the tool means that it can be incorporated into routine pipelines and results can be populated into laboratory information systems using custom scripts, thus avoiding some of the potential errors associated with manual result recording and entry and suiting a role in reference microbiology.

## ACKNOWLEDGEMENTS

We thank McDonald Prest, Doris Omoigui, Tim Chambers and Maimuna Kimuli for retrieving hundreds of isolates from archives and preparing DNA for sequencing. Cath Arnold and the team in Genome Services and Development Unit, PHE Colindale for sequencing all isolates. Ella Campion, Gurkiran Mankoo and John Duncan for performing routine *S. pneumoniae* serotyping and repeat testing some isolates. Steve Platt and Tony

McNiff for LIMS integration developments. Mark Van der Linden, Anni Virolainen-Julkunen and Siira Lotta for sending the 6D pneumococcus strains. Pernille Landsbo Elverdal, and team (Statens Serum Institut) for preparing and sending latex factor sera.

### Funding

The authors received no funding for this work.

### Competing Interests

The authors declare there are no competing interests.

### Author Contributions

- Georgia Kapatai conceived and designed the experiments, performed the experiments, analyzed the data, contributed reagents/materials/analysis tools, wrote the paper, prepared figures and/or tables, reviewed drafts of the paper.
- Carmen L. Sheppard conceived and designed the experiments, performed the experiments, contributed reagents/materials/analysis tools, wrote the paper, reviewed drafts of the paper.
- Ali Al-Shahib contributed reagents/materials/analysis tools, reviewed drafts of the paper.
- David J. Litt reviewed drafts of the paper, lab support.
- Anthony P. Underwood reviewed drafts of the paper, bioinformatics advice.
- Timothy G. Harrison reviewed drafts of the paper, former Unit Head (Retired).
- Norman K. Fry reviewed drafts of the paper, Section Head.

### DNA Deposition

The following information was supplied regarding the deposition of DNA sequences:

Three novel sequences can be found as assemblies at the PHE Pathogens BioProject PRJEB14267 at ENA (http://www.ebi.ac.uk/ena/data/view/PRJEB14267).

### Data Availability

Code repository for PneumoCaT:

https://github.com/phe-bioinformatics/PneumoCaT;

Raw data can be found at the PHE Pathogens BioProject PRJEB14267 at ENA (Data available from Monday 13/06/2016) (http://www.ebi.ac.uk/ena/data/view/PRJEB14267).

### Supplemental Information

Supplemental information for this article can be found online at http://dx.doi.org/10.7717/peerj.2477#supplemental-information.

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
