# Peer review of "Whole genome sequencing of Streptococcus pneumoniae: development, evaluation and verification of targets for serogroup and serotype prediction using an automated pipeline"

_PeerJ, doi:10.7717/peerj.2477_

## Round 0.1 · original submission · Minor Revisions

· Academic Editor

Minor Revisions

I have received two thorough reviews of your paper and both reviewers found it well-written and interesting. I concur and am happy to move forward with your paper pending some minor revisions. Please note especially Reviewer 1's comments with respect to the functionality of the software. This clearly needs to be addressed for the paper to move forward.

·

Basic reporting

This is a very well written manuscript with a clearly defined objective. The authors are presenting an automated tool that predicts Streptococcus pneunomiae serotypes (and serogroup) with higher accuracy than the current phenotypic immunological testing procedure. They walk through the pipeline design, and then validate the results with several large published datasets of phenotypically determined serotypes.

No negative comments.

Experimental design

No negative comments to report. Experimental design was good.

Validity of the findings

The authors present results that drastically improve the repeatability, accuracy, and sensitivity of current methods.

Comments for the author

Overall this manuscript was a pleasure to read. It is well-written and well organized. Most of my comments below are on trying to run the program itself, not on the manuscript.


Major edits

1. Error when running the program on the Example files provided:

user:~/analyses/PneumoCaT$ python PneumoCaT.py -i Examples/PHESPV1910
running bowtie index
There was an error in the function 'output_all'
* * *
Traceback (most recent call last):
File "/home/user/analyses/PneumoCaT/modules/utility_functions.py", line 59, in try_and_except
return function(*parameters, **named_parameters)
File "/home/user/analyses/PneumoCaT/modules/Serotype_determiner_functions.py", line 306, in output_all
qual_ascii = pileupread.alignment.qual[pileupread.qpos]
AttributeError: 'pysam.calignedsegment.PileupRead' object has no attribute 'qpos'
* * *
Traceback (most recent call last):
File "PneumoCaT.py", line 140, in <module>
main(opts)
File "PneumoCaT.py", line 125, in main
hits = Serotype_determiner_functions.find_serotype(opts.input_directory, fastq_files, reference_fasta_file, opts.output_dir, opts.bowtie, opts.samtools, opts.cleanup, id, logger, workflow=workflow, version=version) ## addition for step2
File "/home/user/analyses/PneumoCaT/modules/Serotype_determiner_functions.py", line 57, in find_serotype
hits = try_and_except(input_directory + "/logs/strep_pneumo_serotyping.stderr", output_all,bam,reference_fasta_file,output_file,id,workflow,version) # added for step 2
File "/home/user/analyses/PneumoCaT/modules/utility_functions.py", line 77, in try_and_except
error_file = open(error_filepath, "a")
IOError: [Errno 2] No such file or directory: 'Examples/PHESPV1910/logs/strep_pneumo_serotyping.stderr'

Minor edits

GitHub Documentation:
a. Overall the documentation is very thorough. However, I would highly recommend providing a tutorial section with the exact command on running the program on the example files. The “Examples” section of the documentation only provides suggestions on how to set up your input files. I was not able to successfully run the program on the example files given.

b. Also, provide brief installation instructions for all the dependencies. Even if it’s just the linux command for installing the library, i.e. “pip install pysam”.

c. Dependancies: Include the “2” in Bowtie. Bowtie2. The program doesn’t run with vs. 1.

Text Edits:
d. Line 182: Suggested RAXML model: For analyzing variant-only data you might consider using Paul Lewis’s model correction for ascertainment bias:

raxml-PTHREADS -s snpma.fasta -n OutPutName.tre -m ASC_GTRCAT -v –asc-corr=lewis -p 123 -T 8

• The ASC_GTRCAT is the general model, then the ascertainment correction (–asc-corr) is “Lewis”, from Paul Lewis

e. Line 256: double comma typo

Reviewer 2 ·

Basic reporting

The manuscript titled "Whole genome sequencing of Streptococcus pneumoniae: Development, evaluation and verification of targets for serogroup and serotype prediction using an automated pipeline" by Kapatai et al., is clearly written using professional English. They did a great job in explaning the background and the need for such a typing tool for S. pneumoniae. All the figures are relevant, well described and of high quality, which adheres to PeerJ policy

Experimental design

The experimental design is good and the research statement is well defined. The methods described ( both using mapping of reads and data from denovo contigs + SNP-based variant detection using mapping) are standard methods, which are appropriate for this type of analysis.

Validity of the findings

I have no comments. The conclusions are well stated.

Comments for the author

It is a useful tool for S. pneumoniae community. A general trend that I observe in such community or bug-specific software tools are the lack of continuing support. Due to WGS being a routine practice and more carriage studies being implemented, chances are high that more capsular locus reference sequences ( more than 94 used in this software) will be discovered in the future. Are the authors planning provide continuous support in updating the reference sequence database.
Based on above mentioned concern, I do have 2 suggestions:

1. It would be great if the authors allow the users to specify there own database of reference capsular locus sequences. Please provide instructions on how to generate a custom database and how to specify them in the command line.

2. Many of researchers will have access to denovo assemblies and pacBio data. It would be great to add this component as an input file for the software. I know the read mapping data is more valuable than contigs, but this customization will help more users to use the tool.

---

## Round 0.2 · accepted · Accept

· Academic Editor

Accept

Thank you for your careful revision of your paper. Both the previous reviewers appreciated your responses to their concerns. I am now happy to recommend acceptance of your paper.

·

Basic reporting

All of my reviewer comments were addressed in this revision. Despite having both Bowtie2 and samtools programs included in my PATH variable, I'm able to run the program with the paths given as arguments. Congrats to the authors for a great manuscript.

Experimental design

none

Validity of the findings

none

Comments for the author

none

Reviewer 2 ·

Basic reporting

No Comments

Experimental design

No Comments

Validity of the findings

All the concerns and comments are addressed by the authors.

Comments for the author

I am satisfied with the response given by the authors to my comments.